# Characterization of Bacteriophages against *Pseudomonas Syringae* pv. *Actinidiae* with Potential Use as Natural Antimicrobials in Kiwifruit Plants

**DOI:** 10.3390/microorganisms8070974

**Published:** 2020-06-29

**Authors:** Oriana Flores, Julio Retamales, Mauricio Núñez, Marcela León, Paula Salinas, Ximena Besoain, Carolina Yañez, Roberto Bastías

**Affiliations:** 1Laboratory of Microbiology, Institute of Biology, Pontificia Universidad Católica de Valparaíso, Av. Universidad # 330, Curauma, 2340000 Valparaíso, Chile; oriana.flores@pucv.cl (O.F.); julio.retamales@pucv.cl (J.R.); m.nunez.rodriguez@gmail.com (M.N.); marcela.leon@pucv.cl (M.L.); carolina.yanez@pucv.cl (C.Y.); 2Facultad de Ciencias, Escuela de Biotecnología, Universidad Santo Tomás, 8370003 Santiago, Chile; paulasalinassa@santotomas.cl; 3Laboratory of Phytopathology, School of Agronomy, Pontificia Universidad Católica de Valparaíso, 2260000 Valparaíso, Chile; ximena.besoain@pucv.cl

**Keywords:** bacteriophage, kiwifruit, Psa, *Pseudomonas syringae*, *Pseudomonas syringae* pv. *actinidiae*

## Abstract

*Pseudomonas syringae* pv. *actinidiae* (Psa) is the causal agent of a bacterial canker in kiwifruit plants and has caused economic losses worldwide. Currently, the primary strategies to control this pathogen include the use of copper-based compounds and even antibiotics. However, the emergence of isolates of Psa that are resistant to these agrochemicals has raised the need for new alternatives to control this pathogen. Bacteriophages have been proposed as an alternative to control bacterial infections in agriculture, including Psa. Here, we show the isolation and characterization of 13 phages with the potential to control Psa infections in kiwifruit plants. The phages were characterized according to their host range and restriction fragment length polymorphism (RFLP) pattern. Four phages were selected according to their lytic effect on the bacteria and their tolerance to different environmental conditions of pH (4–7), temperature (4–37 °C), and solar radiation exposure (30 and 60 min). The selected phages (CHF1, CHF7, CHF19, and CHF21) were sequenced, revealing a high identity with the podophage of Psa phiPSA2. In vitro assays with kiwifruit leaf samples demonstrated that the mixture of phages reduced the Psa bacterial load within three hours post-application and was able to reduce the damage index in 50% of cases. Similarly, assays with kiwifruit plants maintained in greenhouse conditions showed that these phages were able to reduce the Psa bacterial load in more than 50% of cases and produced a significant decrease in the damage index of treated plants after 30 days. Finally, none of the selected phages were able to infect the other bacteria present in the natural microbiota of kiwifruit plants. These results show that bacteriophages are an attractive alternative to control Psa infections in kiwifruit plants.

## 1. Introduction

The nutritional value of kiwifruit is associated with a high content of antioxidants, vitamins, flavonoids, and minerals, positioning this fruit as an excellent functional food for the prevention of cardiovascular diseases and even cancer [1,2]. Kiwifruit represents between 0.2 and 0.3% of total fresh fruit production worldwide. Fuzzy kiwifruit (*Actinidia deliciosa*) alone generates 1.8 million tons of production per year. Chile, Italy, and New Zealand are the world’s largest exporters of kiwifruit, and together they represent approximately 80% of the exportations [3]. The intensive production of kiwifruit facilitates the appearance of phytopathogens, such as *Pseudomonas syringae* pv. *actinidiae* (Psa), which is the primary bacterial pathogen in this plant, generating economic losses of approximately 280 million USD [4]. This pathogen is the etiological agent of the bacterial canker in kiwifruit and has a worldwide distribution. Therefore, it is considered a pandemic [5]. According to the genetic and phenotypic analysis of different Psa isolates obtained worldwide, there are six biovars of Psa, of which Biovar 3 is responsible for the current pandemic outbreak [6,7].

During 2014, Chilean kiwifruit production reduced its exports by approximately 55% due to the heavy frosts experienced in the season and due to an outbreak of Psa [8]. The control methods against Psa include the adoption of Good Agricultural Practices (GAP) to reduce the proliferation of outbreaks, but, most importantly, the use of bactericidal compounds, such as copper and antibiotic formulations [9]. However, the appearance of Psa strains resistant to copper and antibiotics [10,11,12], together with the evidence of phytotoxicity caused by these compounds [13], has led to the search for new alternatives to control this pathogen. 

Biocontrol strategies, such as the use of antagonistic bacteria, like *Bacillus subtilis, Pantoea agglomerans*, and *Lactobacillus plantarum*, have shown promising results against Psa in in vitro conditions [14,15,16]; however, the efficacy of this approach to control Psa infection in commercial field conditions was less consistent [15,17,18,19]. Bacteriophages have also been assayed as controlling agents against phytopathogens. Their specificity to kill the host bacteria and, in consequence, a reduced probability of producing dysbiosis on plants, and their non-toxic and self-replicating nature situate phages in a better position over antibiotics to control pathogenic bacteria in plants [20,21,22].

Several bacteriophages against different pathovars of *Pseudomonas syringae* have been reported [23,24,25]. Phages against *P. syringae* pv. *actinidiae* have been isolated in Italy [26], New Zealand [27,28,29], Korea [30] and China [31]. All these phages have been subjected to general characterization, showing the potential to control Psa in in vitro conditions. More recently, a study from Portugal also demonstrated that the phage phi6, obtained from the DSMZ collection (DSM 21518), has the potential to reduce the bacterial concentration of different Psa strains in ex vivo tests of kiwifruit tree leaves in laboratory conditions [32].

In this study, we show the characterization of four phages with the potential to control Psa in vitro and in vivo under greenhouse conditions, indicating that bacteriophages are a promising alternative for the biocontrol of Psa, to reduce the symptoms and bacterial load in kiwifruit plants.

## 2. Materials and Methods

### 2.1. Bacterial Strains and Culture Conditions

*Pseudomonas syringae* pv. *actinidiae* (Psa) Chilean isolates were obtained from the Agricultural and Livestock Service (SAG) and characterized in a previous study [33]. The Psa strains used for phage isolation were Psa889, Psa386, Psa189, Psa233, and Psa381. The specificity of phages was tested with the bacterial strains *Pseudomonas putida* KT2442, *Pseudomonas aeruginosa* PAO1, *Pseudomonas antartica* S63, and the commensal kiwifruit bacterial isolates *Exiguobacterium* sp., *Bacillus* sp., and *Pantoea* sp. All the strains belonged to the bacterial collection of the Laboratory of Microbiology of the Pontificia Universidad Católica de Valparaíso (PUCV). The bacteria were grown at 25 °C in liquid and solid (1.5% agar) lysogeny broth (LB) medium at 180 rpm when necessary. A Double-agar assay was performed as described previously [34]. Briefly, 500 µL of an early exponential bacterial culture (OD_600_ nm = 0.2) was added to 3 mL of soft agar (LB, 0.6% agar) maintained at 50 °C. After gentle mixing, the culture was poured onto LB-agar plates to form a bacterial top-agar layer, left to solidify at room temperature, and incubated at 25 °C for 18 h.

### 2.2. Bacteriophage Isolation, Propagation, and Concentration

Soil samples were facilitated by SAG, and obtained from kiwi orchards that were positive for Psa. Water samples (creeks and lagoons with different locations) were collected from the central–southern zone of Chile (Regions of Valparaíso, O’Higgins, Maule, and Metropolitana). The samples for phage isolation were treated as follows: 5 g of soil samples were mixed with 45 mL of PBS buffer (137 mM NaCl, 3 mM KCl, 7 mM Na_2_HPO_4_, and 1.5 mM KH_2_PO_4_, at pH 7.2–7.4) and centrifuged (10,000× *g* for 10 min at 4 °C). The supernatant (5 mL) was mixed with 30 mL of the corresponding Psa culture and incubated at 25 °C. 

In the case of the water samples, 300 mL was mixed with 30 mL of a Psa exponential culture and incubated at 25 °C for enrichment. The debris was removed by centrifugation, and 100 µL of the supernatant was added to a Psa soft-agar lawn. Lytic plaques that displayed unique morphology were isolated three times using the streaking method to assure the isolation of unique phage clones. Phage propagation was performed by suspending a lytic plaque in 1 mL SM buffer (10 mM Tris–HCl, pH 7.5; 100 mM NaCl; and 10 mM MgSO_4_), which was then mixed with 100 mL of the corresponding Psa culture (OD 600 nm = 0.3) and incubated overnight at 25 °C. The lysates were centrifuged (10,000× *g* for 30 min at 4 °C), and the filtered supernatant (a 0.22-µm-pore-size filter) was stored at 4 °C. The phage particles from the lysates were precipitated with 10% (*w*/*v*) polyethylene glycol (PEG 8000) in 1 M NaCl at 4 °C for 10 h and then centrifuged (10,000× *g*, 10 min, 4 °C). The phage pellets were suspended in 500 µL of SM buffer, and chloroform was added at 1% *w/w* proportion. The phage preparations were then vortexed and centrifuged (13,000× *g* for 10 min at 4 °C), and the supernatant was stored at 4 °C until use. The phage titer (PFU/mL) was determined using a standard double layer agar assay as described previously [34].

### 2.3. Bacteriophage Viability under Different Conditions

To determine the survival of the phages under different temperatures, each phage suspended in SM buffer was incubated at 4, 18, and 37 °C for 1 h. For survival under different pH conditions, the phages were suspended in 1 mL SM buffer, previously adjusted with 1 M NaOH or 1 M HCl, to yield a pH range from 4.0 to 6.0 and then incubated at 25 °C for 1 h. In the case of survival under solar radiation, phages in SM buffer were exposed to the sun during a day with a solar UV index of level 8 for 30–60 min. After incubation in each condition, serial dilutions of the treated phage samples were titered with the corresponding Psa host in a double-layer-agar assay. All the assays were carried out in biological triplicate.

### 2.4. Molecular and Morphological Characterization of Bacteriophages

Genomic bacteriophage DNA extraction was performed with a phage DNA isolation kit (Norgen Biotek Corp., Thorold, ON, Canada) according to the manufacturer’s instructions. Restriction fragment length polymorphism (RFLP) analysis was performed by digesting the phage DNA with the enzymes *HhaI* and *MspI* (NEB) according to the manufacturer’s instructions. The restriction patterns were visualized in bisacrylamide:acrylamide gels (12%) with silver staining. The restriction patterns were analyzed using the software PyElph [35]. The phage genomes were sequenced on Illumina Hiseq 2000 (Illumina Inc., San Diego, CA, USA) at Macrogen Inc. (Seoul, Korea). The reads were processed for the removal of sequencing adapters, quality trimming (Q > 30), and assembling of paired-end reads using Geneious R11 software (www.geneious.com) [36].

The phages were assembled following a de novo strategy and the results were then confirmed performing a new assembling using the phage phiPSA2 as a reference genome [26]. Contigs of putative assembled phage genomes were identified based on the high coverage by PHAge Search Tool—Enhanced Release (PHASTER, http://phaster.ca/) [37]. Phage ORFs were identified by Geneious and PHASTER [36,37] and were annotated according to the maximum identity determined by BLASTP (NCBI). The accession numbers of the phage genomes in GeneBank (NCBI) were CHF1 (MN729595), CHF7 (MN729596), CHF17 (MN729600), CHF19 (MN729597), CHF21 (MN729598) and CHF33 (MN729599). Nucleotide alignment between the phage genomes was done using the Mauve tool [38], and genome alignment representation was done with BLASTn and EasyFig [39]. The morphology of the phages was determined through transmission electron microscopy (Phillips Tecnai 12, Biotwin), and they were negatively stained with 2% aqueous uranyl acetate (pH 4.0). Images were captured with a SIS CCD camera Megaview (Olympus Corp., Tokyo, Japan) in the Advanced Microscopy Unity of Pontificia Universidad Católica de Chile.

### 2.5. Infection Curves, Bacteriophage Host Range Analysis, and Resistance Frequency

Infection curves were performed by adding 200 µL of each phage preparation to the corresponding early exponential Psa culture (10^3^ CFU/mL) using the multiplicity of infections (MOIs) of 0.1, one, and 10. Infection curves were performed in 96 multi-well plates at 25 °C over 20 h in a microplate spectrophotometer Infinite RM200 NanoQuant (TECAN). The OD (600 nm) was taken every 30 min. The corresponding Psa host cultures without phages were used as controls. The host range determination was performed though a spot test and a standard double layer assay [34]. The lytic activity of the phages was classified into four categories: strong activity (complete clear lysis plaque); medium activity (clear throughout but with faintly hazy background lysis plaque); light activity (turbid lysis plaque); and no activity (no lysis plaque observed). To obtain the bacterial resistance frequency, a culture of the bacterial host at OD 0.1 (~10^7^ CFU/mL) was infected with the corresponding phage (MOI > 100). After 10 min of incubation, serial dilutions of the cultures were plated in a solid medium previously inoculated with phages; therefore, only resistant bacteria will be able to grow. In parallel, bacterial cultures inoculated with an equal volume of fresh medium were used as controls. After 48 h of incubation, the frequency of resistance was obtained by calculating the ratio between the concentration of the corresponding Psa strain grown in the presence or absence of the phage [40]. All assays were performed in biological and technical triplicate.

### 2.6. Efficacy of Bacteriophage Cocktail in Psa Control on Kiwifruit

In vitro assays to evaluate the effect of phages in the Psa load were performed with leaf discs (2 cm diameter). The discs were obtained from healthy kiwifruit plants (*Actinidia chinensis* var. deliciosa “Hayward”). The surfaces of the discs were disinfected with sodium hypochlorite 1% and washed twice with sterile distilled water. Subsequently, the discs were placed in humidity chambers and inoculated individually with three drops of 10 µL of Psa743 isolate (1 × 10^6^ CFU/mL). The phage cocktail was added two hours after Psa with a MOI of 10. Leaf discs only infected with Psa were used as a control group. The Psa bacterial load was determined by homogenizing leaf disc samples in 1 mL of magnesium buffer (10 mM MgSO4) following serial dilutions and plating on a solid medium for the determination of the Psa concentration (CFU/mL). The quantification of bacteriophages (PFU/mL) was performed with a double-agar layer plaque assay with a Psa 743 isolate as the indicator strain.

The assays to evaluate the effect of phages in the damage produced by Psa over the leaf discs were performed according to the protocol described by Prencipe et al. (2018) [41], with some modifications. Briefly, a disinfected kiwifruit leaf disc was deposited in a six-well plate containing 10 mL of sterile distilled water supplemented with cycloheximide (100 μg/mL) to avoid fungus growth. The leaf discs were infected with three equidistant drops of 15 μL of Psa 743 suspended in magnesium buffer (10 mM MgSO4) (1 × 10^8^ CFU/mL). After two hours, 15 μL of the phage cocktail suspension (MOI = 10) was added directly on the leaf discs, then they were incubated at 20 °C. In the corresponding cases, the phage cocktail was inoculated again at 24, 48, and 72 h. Damage in the leaf discs was evaluated after three days using the Disease Index (DI) (0–4) scale proposed by Prencipe et al. 2018 [41]. All experiments were performed in biological and technical triplicate, evaluating a total of 18 discs for each condition. Leaf discs inoculated with sterile deionized water were used as a control. Statistical analysis was performed using one-way ANOVA and Tukey’s multiple comparisons test with *p* ≤ 0.05.

The in vivo experiments were performed with two-year old kiwifruit plants (*A. chinensis var. deliciosa* “Hayward”) maintained under greenhouse-controlled conditions (20 °C ± 2 and a natural photoperiod). To evaluate the effect of the phages in the reduction in the Psa load, the plants were infected with a bacterial suspension of Psa in magnesium buffer (10^6^ CFU/mL) by spraying the suspension directly over the surface of the plant leaves. The relative humidity was raised to approximately 100% for up to 72 h after inoculation to promote Psa invasion and then reduced to near 70% [42]. After 2 h post-infection (hpi), the cocktail of phages was added to the plants with a MOI of 10 by spraying. A second dose of the cocktail of phages was added 20 hpi in the indicated cases. After 24 h, the Psa bacterial load was determined from 1 g of leaves macerated in 1 mL magnesium buffer, which was then used for serial dilutions determination of Psa concentration (CFU/mL).

Two independent experiments were performed to evaluate the biocontrol effect of phages in kiwifruit plants infected with Psa. Each experiment included five plants per treatment and five infected leaves per plant, evaluating a total of 50 leaves for each treatment. In this case, the plants were infected as described above but with a mixture of three Psa strains (Psa 743, Psa 889, and Psa 598) (10^8^ CFU/mL/leaf). The phages were added, as described above, 1 h post-infection using a MOI of 10 (10^8^ PFU/mL/leaf). Copper sulfate was added following the dose and recommendations of the manufacturer, and plants not infected with Psa were used as a control. All the plants were properly labeled and then randomly distributed in the greenhouse. After 30 days, the damage index on leaves (0–5) was assigned according to the scale proposed by Prencipe et al. 2018 [41], and the Psa load was determined as described above. Statistical analysis was performed using one-way ANOVA and Tukey’s multiple comparisons test with *p* ≤ 0.05.

## 3. Results

### 3.1. Isolation and Selection of Bacteriophages against Pseudomonas syringae pv. actinidiae

The phages were isolated using different strains of Psa biovar 3 obtained from Chilean kiwifruit orchards as the host [33]. Fourteen bacteriophages were isolated from soil and water samples collected from agricultural regions of Central-South Chile. Bacteriophages were named as vB_PsaP-CHF and a number corresponding to the sample number. Throughout the document, the phages will be referred to as CHF and the corresponding number (see Appendix A). All bacteriophages have a double-stranded DNA genome, and an RFLP analysis with the enzymes *HhaI* and *MspI* revealed very similar patterns between them, distinguishing only slight differences on specific bands (Appendix A). Similarly, a host range assay revealed that all bacteriophages were able to infect the 18 different Psa isolates included in this study, and the phages CHF21, CHF30, and CHF33 were observed with more lytic activity against all the isolates (Appendix A).

Considering the potential application of the phages as antimicrobials against Psa in kiwifruit orchards, the isolated phages were exposed to different environmental conditions of pH, temperature, and solar radiation to evaluate their survival. The results show that the phages demonstrated different tolerances to acidic conditions, with some phages, such as CHF4, CHF15, and CHF16, showing a significant reduction by more than two orders of magnitude in their titer after 1 h of exposure to pH 4 and 5, while other phages, such as CHF1 and CHF7, were able to endure these conditions (Figure 1A).

The survival of the phages was also evaluated after 1 h of incubation at 37, 18, and 4 °C. In the majority of cases, the survival of the phages was not affected under these conditions. However, at 37 °C, the phages CHF1 and CHF30 showed a reduction in their titer in more than one order of magnitude to 1.5 × 10^5^ PFU/mL and 1.4 × 10^5^ PFU/mL, respectively (Figure 1B). The phages were not exposed to higher temperatures because they are not commonly expected during the Psa infection period. Finally, most of the phages showed a reduction in their titer after exposure to solar radiation (UV radiation level 8), which increased with the exposure time (Figure 1C). The phages CHF4 and CHF18 appeared to be the most affected, showing a reduction in their titer in more than one order of magnitude to 4 × 10^4^ PFU/mL for the first and 1.2 × 10^4^ PFU/mL the second, after 30 min of exposition. During the same period, the phages CHF7 and CHF30 did not show a significant reduction in their titer. However, after 60 min of exposure, their titer decreased to 1.9 × 10^3^ PFU/mL and 3.7 × 10^4^ PFU/mL, respectively.

Interestingly, some phages, such as CHF17 and CHF 19, among others, showed no differences in their titer between 30 and 60 min of exposition to solar radiation. These results indicate that all the phages retained activity after exposure to conditions that can be associated with kiwifruit production; however, some of them presented more tolerance than others. After consideration of the results obtained at this point, the bacteriophages CHF1, CHF7, CHF17, CHF19, CHF21, and CHF33 were selected for further characterization and to evaluate their effectiveness for the biocontrol of Psa through in vitro and in vivo assays in greenhouse conditions.

### 3.2. Genomic and Phenotypic Characterization of Selected Psa Bacteriophages

All the selected phages produced large clear lytic plaques in Psa lawns, but occasionally small size plaques were also observed (Figure 2A) (CHF1, CHF7, CHF17, CHF19, CHF21, and CHF33). All phages belong to the Podoviridae family according to their virion morphology, with a head of 60 ± 0.6 nm of diameter (Figure 2B). The genome of the selected phages was sequenced using NGS with a sequence coverage above 1000 for each phage. The phages have a genome size ranging between 40,557 and 40,999 bp, flanked by direct terminal repeats (DTRs) of 216 bp and a GC% near 57% (Figure 3, Appendix A). An alignment between the genomes showed that these phages shared at least 96.6% nucleotide identity. The phages CHF1 with CHF33, and CHF19 with CHF17 shared nucleotide identity of 99.96% and 99.99%, respectively (Appendix A). This analysis showed that all the sequenced phages were very close relatives and that the phages CHF1 with CHF33, and CHF19 with CHF17 were essentially the same phage. The phage CHF33 showed stronger lytic activity over different Psa isolates in comparison to CHF1 (18 vs. 14 isolates); however, the titers obtained during phage production were consistently higher in CHF1 and CHF19 than CHF33 and CHF17. Therefore, the latter phages were excluded from further experiments. The combination of the selected phages CHF1, CHF7, CHF19, and CHF21 demonstrated efficient infections in all the Psa isolates tested in the study (Appendix A).

The annotation of the genomes shows that all the Chilean Psa phages had a genome organization similar to T7-like phages, and similar to the Psa bacteriophage, phiPSA2, isolated in Italy [26], sharing identities ranging from 93.2% to 96.4% (Figure 3, Appendix A). In total, 48 ORFs were identified, and no genes coding for virulence factors or antibiotic resistance were found (Appendix A). Only the phages CHF1 and CHF33 contained an extra ORF (ORF 0), which was annotated as a hypothetical protein. In spite of the high genome identity between the phages, the analysis revealed differences in SNPs, and deletions/insertions were distributed in different regions of the genomes. These differences were found in the genes of RNA polymerase, DNA ligase, lysozyme, primase/helicase, DNA polymerase, tail fiber proteins, internal virion proteins, and hypothetical proteins, among others (Figure 3). Principal differences between the Psa phages proteins were detected in RNA polymerase (98.4% identity) and Internal virion protein D (97.9% identity) The genomic analysis suggested that these phages are closely related and to the phage phiPSA2. Whether these phages are the same phage or same species is a matter that surpasses the aims of this study and will be addressed in the discussion section (Section 4).

### 3.3. In Vitro Efficacy of Bacteriophages to Control Psa

The lytic activity of the phages CHF1, CHF7, CHF19, and CHF21 was evaluated through infection curves using the respective Psa host for each phage. The results show that all phages were able to clear the bacterial culture, even when a MOI of 0.1 was used (Figure 2C). The phage CHF1 was the most effective at lysing the bacterial culture within 2 h, while the phage CHF19 needed up to 6 h to clear the Psa culture. Considering the potential use of these phages as biocontrol agents against Psa, the frequency of bacteriophage resistant mutants was evaluated. The phages CHF1, CHF7, CHF19, and CHF21 selected for bacteriophage resistant mutants with a frequency of 4.79 × 10^−6^, 9.05 × 10^−6^, 9.3 × 10^−6^, and 3.28 × 10^−6^, respectively. In all cases, three different colony morphologies were observed (normal Psa-like, small, and smooth borders (data not shown)).

The potential of these phages as biocontrol agents for Psa was first evaluated under laboratory conditions with kiwifruit leaf samples. Our results show that a cocktail of phages CHF1, CHF7, CHF19, and CHF21 in equal proportion was able to reduce the bacterial load of Psa over leaves below our detection limit (20 UFC/mL) within 3 h post-infection (hpi), and remained undetected up to 24 hpi (Figure 4A). During the same period, the Psa load in the untreated leaves ranged between 10^5^–10^7^ UFC/mL. Bacteriophages were detectable throughout the entire experiment, showing an increase in their titer at 3 h, probably due to their replication in Psa (Figure 4A).

To evaluate if the cocktail of phages was able to protect kiwifruit leaves against the damage produced by Psa, an in vitro methodology proposed by Prencipe et al. (2018) [41] was implemented. The results show that these phages were also able to protect kiwifruit leaf discs from the damage produced by Psa. After three days, leaf discs that received the phage cocktail treatment presented an average damage index of 1.5, while the untreated discs presented a damage index of 2.9. Multiple doses of phage cocktail resulted in a damage index of 0.8, showing significant differences with the untreated discs (Figure 4B). No damage was observed in the control leaf disc groups that were not infected with Psa or treated with phages. These results suggest that this cocktail of bacteriophages had the potential to control the infection produced by Psa in kiwifruit tissue, reducing the bacterial load and symptomatology in leaves.

### 3.4. Specificity of Chilean Psa Bacteriophages

One of the advantages of bacteriophages over other antimicrobials used in agriculture is their specificity. Therefore, the lytic activity of the selected phages was tested against other bacterial species. Due to governmental regulations and the quarantine status of Psa in Chile, it was not possible to evaluate if these phages were able to infect other pathovars of *Pseudomonas syringae*. However, we did test if they were capable of infecting other bacteria isolated from healthy kiwifruit plants, together with other *Pseudomonas* species. None of these phages were able to infect any of the 15 isolated bacteria, including *Exiguobacterium* sp., *Bacillus* sp., and *Pantoea sp*, and neither *Pseudomonas putida KT2442*, *Pseudomonas aeruginosa PAO1*, or *Pseudomonas antartica S63*, suggesting that no native bacteria would be affected by these phages.

### 3.5. In Vivo Efficacy of Bacteriophages to Biocontrol Psa under Greenhouse Conditions

Two-year old Kiwifruit plants (*A. chinensis var. deliciosa* “Hayward”) maintained in greenhouse conditions were used to determine the potential of these phages to control Psa in vivo. The results show that the cocktail of phages was able to reduce the Psa load over kiwifruit leaves by more than 75% in comparison with the untreated plants after 24 h post-infection with Psa (Figure 5A). Despite this bacterial reduction, no significant differences were observed with simple or double-phage applications on the leaf surfaces.

To evaluate if the cocktail of phages was able to combat the disease produced by Psa in kiwifruit plants, two independent assays were performed. The assay also included a commercial copper sulfate used to control Psa as a comparison parameter. In a similar manner to the in vitro assays, after 30 days post-infection, the damage index of plant leaves infected with Psa, and not treated with phages, was 2.3. On the other hand, leaves that did receive the phage cocktail had a damage index of only 1.3 on average (Figure 5B). The symptoms in plants that received copper sulfate treatment presented a damage index of 2.06, similarly to that observed in plants that did not receive any antimicrobial treatment. 

At the end of the assay, the bacterial Psa load was also evaluated, showing that the phage treated leaves presented more than a 70% reduction in Psa in comparison to the untreated leaves; however, these differences were not significant due to the high variability in the bacterial count between different leaves (Appendix A). Conversely, with the observed symptomatology, the leaves treated with copper sulfate showed a strong reduction in their Psa load in comparison to the untreated plants and the phage treated plants. These differences will be discussed later. The joint analysis of these results confirmed the potential of these bacteriophages to control Psa infections in kiwifruit plants.

## 4. Discussion

This study presents the isolation and characterization of phages of Psa. The phages showed different degrees of tolerance to environmental conditions that can be associated with kiwifruit production and were able to infect different Chilean Psa isolates. The selected phages were very efficient at lysing bacterial cultures of this phytopathogen and exhibited great potential to control Psa infection in vitro and in planta when maintained under greenhouse conditions.

### 4.1. Phages against Psa

Initially, 13 phages were isolated, from which six were selected (CHF1, CHF7, CHF17, CHF19, CHF21, and CHF33) according to their host range, genotypification, and survival under different environmental conditions for further characterization and sequencing. The results show that the six phages belonged to the *Podoviridae* family. The sequenced genomes revealed that these phages shared nucleotide identities over 96% among them, with two couples (CHF1 and CHF33; and CHF17 and CHF19) that were essentially identical (>99.9% identity). This information correlates with the observations in the RFLP analysis (Appendix A), where CHF1 share more similitude with CHF33 and CHF17 with CHF19. According to the recommendations of the International Committee of Taxonomy of Viruses (ICTV), these phages corresponded to variants of the same species [43]. The similar identity between these six phages could be explained by the high homogeneity observed among the Chilean Psa strains used as a host [33], although different bacterial isolates were used. A similar situation can be observed at a global scale as the current Psa pandemic is caused by biovar 3 of this phytopathogen [6,44] and the Chilean phages have homology with the phage phiPSA2 isolated in Italy [26], which is also homologous to the phage PPPL-1 isolated in Korea [30,45], and the phage phiPSA17 from New Zealand [27]. This information suggests that, as Psa biovar 3, its phages also have a global distribution.

In this context, it is not correct to claim that the mixture of these phages corresponds to a cocktail of different phages, but variants of the same phage species. Unfortunately, the information regarding the phage sequences was obtained in parallel with the in vitro and in vivo assays, restricting the chance of elaborating a more diverse cocktail. However, despite the high genomic identity of these phages, it is interesting to note the differences in their biological properties, such as the time needed to lyse the bacterial culture (Figure 2C) or different tolerances to environmental conditions (Figure 1). A similar phenomenon was previously described, in which phages with single nucleotide polymorphisms (SNPs) presented remarkable biological differences [46]. More studies would be needed to determine whether the mixture of the phages CHF1, CHF7, CHF19, and CHF21 has the properties of a phage cocktail, despite the high genome identity.

### 4.2. Phages as Biological Control Agents against Psa

To date, there are several reports of the experimental use of phages to control phytopathogens [20,21,47], and some companies such as *OmniLytics* (Omnilytics Inc., Sandy, UT, USA) or *Enviroinvest* (www.enviroinvest.net) are already at the commercialization stage of their phage-based products for agriculture. In all cases, the use of phages in the field is associated with different challenges, such as the tolerance to environmental conditions, including the soil pH, UV radiation, and temperature [48]. The results presented in this study show that all the phages retained their activity after 1 h of exposure to the different environmental conditions associated with kiwifruit production; however, they displayed different degrees of tolerance (Figure 1). These results were considered for the selection of phages used in the assays of biocontrol with kiwifruit leaves and plants.

According to the results, the selected phages were not able to infect other bacterial species present in kiwifruit plants grown in the greenhouse, nor other *Pseudomonas* species. However, other authors have shown that phages against Psa also infected other pathovars of *P. syringae*, such as pv. *morsprunorum* [25], pv. *tabaci*, pv. *tomato*, pv. *phaseolicola* [31], and even *P. fluorescens* [24]; therefore, Chilean Psa phages could possibly infect other *P. syringae* strains and pathovars as well.

Other Psa phages have shown the potential to eliminate Psa from bacterial cultures in laboratory conditions [26,27]. A study from Portugal reported that the phage phi6 was able to reduce the bacterial concentration of Psa in ex vivo tests over kiwifruit leaves in laboratory conditions [32]. In this regard, our study contributes to the general idea of using phages to control Psa, providing evidence that phages can prevent the damage produced by this phytopathogen in kiwifruit plants. The application of phages proved to be effective in reducing the Psa load and damage in leaf tissues through laboratory assays, also showing promising results regarding their effectiveness in combating the bacterial canker in kiwifruit plants.

The assays with phages did not include any excipients or formulations that would extend the survival of phages in the plant, which were undetectable after 48 h post-application. The short viability of phages on kiwifruit leaves could be an explanation of why leaves treated with copper sulfate had a lower Psa load than untreated plants and phage-treated plants. In contrast to the results observed with phages, copper remained on the leaves under greenhouse conditions, where the plants were not impacted by rain. In this scenario, the protective effect of the phages is restricted to the first steps of bacterial infection, likely before the plant invasion.

The development of formulations to increase the persistence of phages on leaf surfaces is a topic that has been explored before [47,49] and should be studied with the Chilean Psa phages, together with an evaluation of their effectiveness in orchards under real kiwifruit production conditions. In this regard, it is also essential to analyze the potential interactions (or interferences) between phages and other agrochemicals commonly used in agriculture, such as fertilizers, or even copper-based compounds [47].

In summary, the renewed attention being paid to phages for the control of pathogenic bacteria, and phytopathogens in particular, raises new challenges in relation to these topics [21]. Our results show that phages have a great potential to control Psa infections in kiwifruit plants. However, more studies are needed to provide a real alternative to control the Psa infections that are currently affecting the kiwifruit production industry.

## Figures and Tables

**Figure 1 microorganisms-08-00974-f001:**
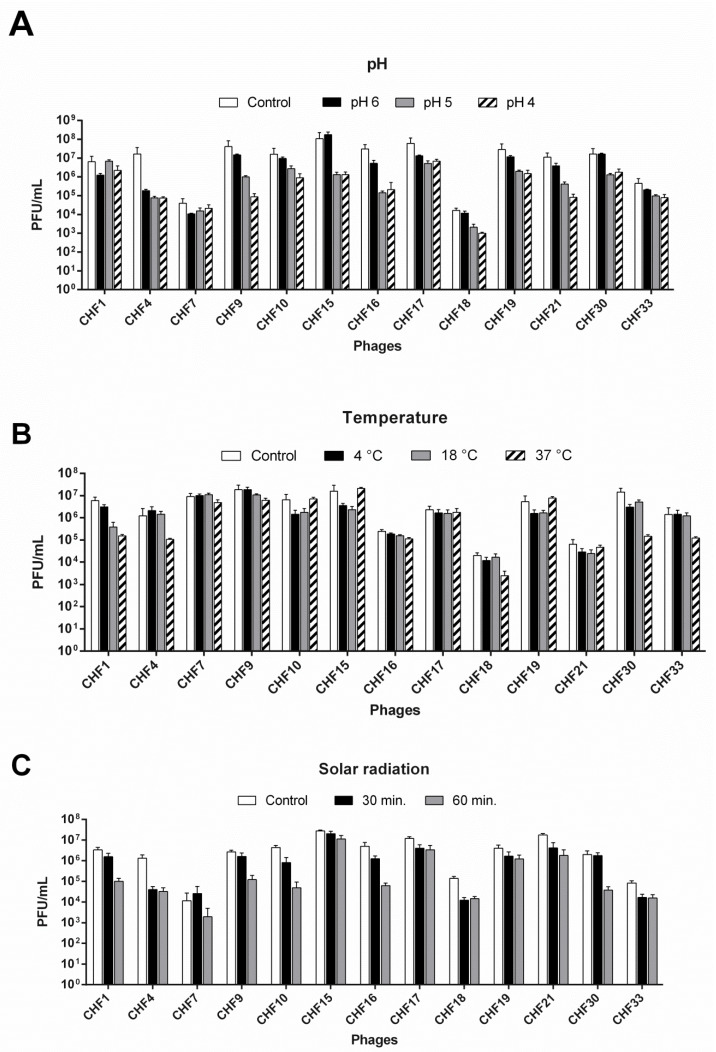
Survival of Chilean *Pseudomonas syringae* pv. *actinidiae* (Psa) phages under different environmental conditions. The fourteen phages were exposed to pH 4–6 for 1 h (**A**), temperatures of 4, 18, and 37 °C for 1 h (**B**) and solar radiation (UV level 8) for 30 or 60 min (**C**). The phage titer was determined in each case and compared with the control condition: Incubation in pH 7 at 25 °C for 1 h without exposition to solar radiation. Experiments were performed in biological triplicate and the standard deviation bars are shown.

**Figure 2 microorganisms-08-00974-f002:**
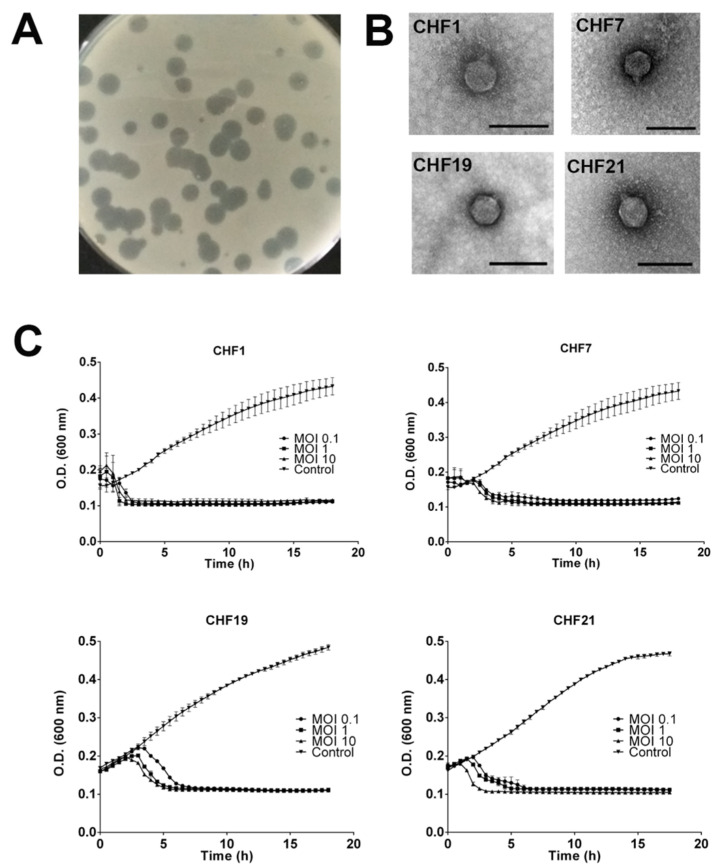
The morphology and lytic activity of selected Psa bacteriophages. (**A**) Lytic plaques of the phage CHF1 in a lawn of Psa strain 889. The rest of the selected phages presented the same lytic plaque morphology. (**B**) Transmission electron micrographs of the phages chosen. Scale bar, 100 nm. (**C**) Infection curves with the different selected phages and the corresponding Psa host with different multiplicity of infections (MOIs) (0.1, 1, and 10). The bacterial population was followed by OD (600 nm) determinations for 18 h. The Psa strain 889 was used for phages CHF1, CHF7, and CHF21, and the strain Psa 233 for phage CHF19. The controls corresponded to the bacterial cultures without the addition of phages. The experiments were performed in triplicate and standard deviation bars are shown.

**Figure 3 microorganisms-08-00974-f003:**
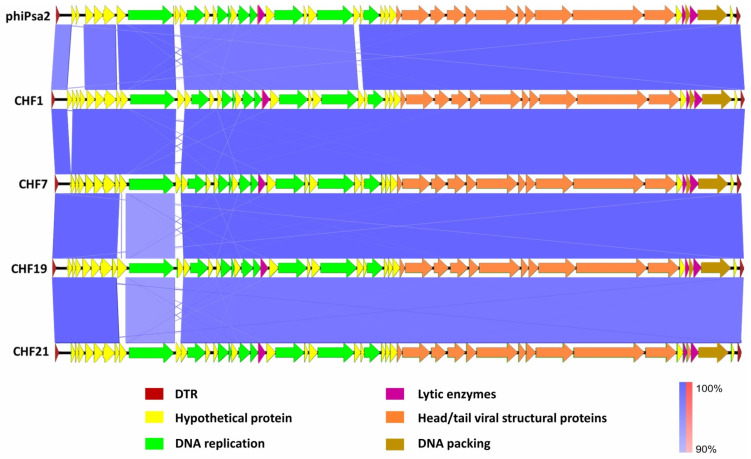
Genomes of phages CHF1, CHF7, CHF19, CHF21, and phiPsa2. The genomes were aligned with BLASTn using Easyfig. Arrows represent the locations of the genes and the vertical blocks the between sequences indicate the level of homology between each phage shaded according to BLASTn. The identity color scale is indicated with blue for matches in the same direction and red for inverted matches. Direct repeat regions (DTR) and predicted genes are labeled in colors according to their predicted function.

**Figure 4 microorganisms-08-00974-f004:**
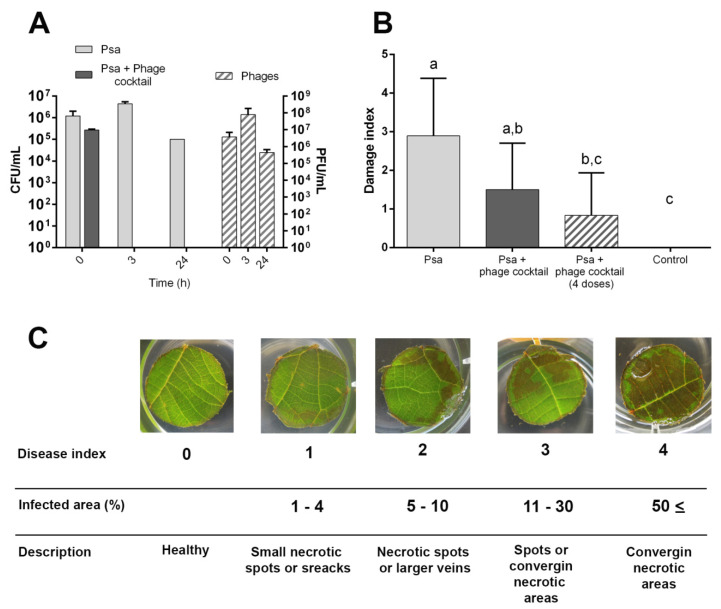
Efficacy of Chilean Psa phages to control Psa in vitro. (**A**) The bacterial Psa load present in leaf discs inoculated with Psa (Light gray) or Psa + phage cocktail (dark gray) after 24 hpi incubation in humidity chambers (*n* = 3). Striped bars represent the phage concentration in leaf discs inoculated with the Psa + phage cocktail. (**B**) The damage index in kiwifruit leaf discs 3 days post-infection with Psa or with the Psa + phage cocktail. Phages were added with a MOI of 10, 1 h (Psa + phage cocktail) or 1, 24, 48, and 72 h (Psa + phage cocktail (4 × doses)) post-infection with Psa. Leaf discs not infected with Psa were used as the control. Phage cocktail: CHF1, CHF7, CHF19, and CHF21. Standard deviation bars are shown. The letters indicate significant differences using one-way ANOVA and Tukey’s multiple comparisons test (*p* ≤ 0.05) (*n* = 18). (**C**) The visual pattern of the damage index used to evaluate the experiment.

**Figure 5 microorganisms-08-00974-f005:**
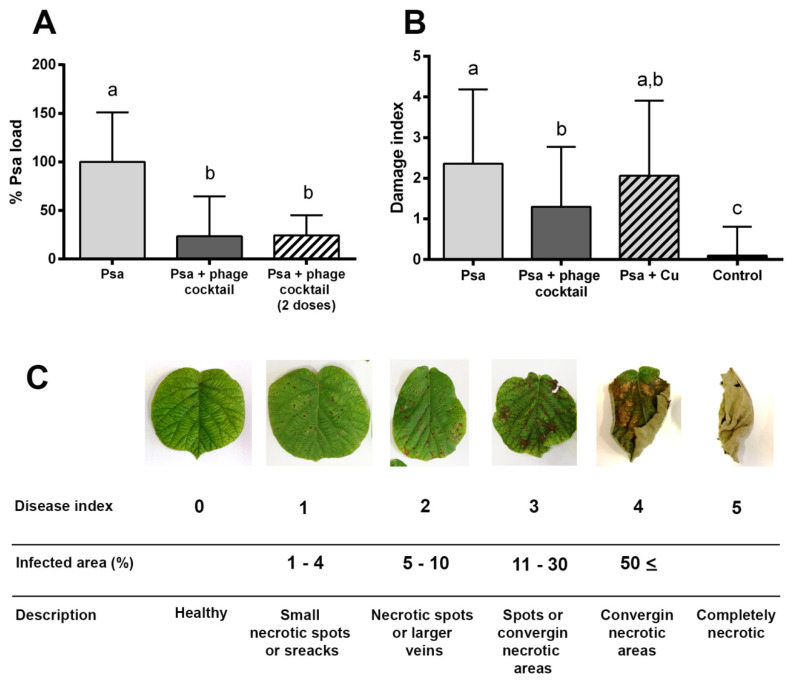
Efficacy of Chilean Psa phages to control Psa in vivo. (**A**) Bacterial Psa load reduction in kiwifruit leaves treated with phages 24 h post-infection with Psa. Phages were added 1 h (Psa + phage cocktail) or 1 and 24 h (Psa + phage cocktail (two doses)) post-infection with Psa. (*n* = 5 leaves per condition). (**B**) Average damage index of two independent experiments performed with kiwifruit plants (*A. chinensis var. deliciosa* “Hayward”) maintained in greenhouse conditions. Kiwifruit plants were infected with Psa alone (light grey), infected with Psa and treated with phage cocktail (dark grey), or infected with Psa and treated with copper sulfate (Cu) (striped bars). The control plants were uninfected and were not treated with phages or copper. The phage cocktail corresponded to an equal mixture of CHF1, CHF7, CHF19, and CHF21 and was added once with a MOI of 10, 1 h post-infection with Psa. The symptomatology in kiwifruit plants was evaluated 30 days post-infection. Five plants for each treatment and five leaves per plant were used in each experiment. Standard deviation bars are shown. The letters indicate significant differences using one-way ANOVA and Tukey’s multiple comparisons test (*p* ≤ 0.05) (*n* = 50). (**C**) The visual pattern of the damage index used to evaluate the experiment.

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
