# Peer review of "Characterization of Bacteriophages against *Pseudomonas Syringae* pv. *Actinidiae* with Potential Use as Natural Antimicrobials in Kiwifruit Plants"

_microorganisms, 2020, doi:10.3390/microorganisms8070974_

Round 1

Reviewer 1 Report

This manuscript describes the isolation, characterisation, in vitro and in vivo activity of phages against the kiwifruit pathogen, Pseudomonas syringae pv. actinidiae (Psa), responsible for the current pandemic. The results are interesting and significant given the current impact of this bacteria on the agricultural field, although, there are several comments that need to be addressed. A thorough characterisation of the phages is included (host range, specificity, stability, TEM and WGS, etc.), however, the in vitro and in vivo results sections need clarification to avoid misinterpretation (see comments). Similarly, there are several grammatical errors that require correction and proof-reading of the entire manuscript.

Specific comments:

  • Line 116: Please name Restriction Fragment Length Polymorphism (RFLP) in full before referring to the acronym.
  • Figure 1: There are a few corrections that would improve the readability of this figure. Firstly, the in-figure legend should read in the same order as the columns are presented as in panel A. Panel B and C are written in the reverse order with control last. In addition, it would be beneficial to use a more discerning colour scheme to easily visualise the different conditions.
  • Line 219, Figure 1 description: Reword this for clarity to something like “…phages were exposed to pH 4 – 6 for 1 hour (A), temperature of 4°, 18° and 37°C for 1 hour and solar…”.
  • Line 238-240: It is interesting that of the near identical phages CHF1 and CHF33 that the latter (CHF33) was excluded from further experiments given that the host range data suggests greater lytic activity compared to CHF1 against Psa hosts 598, 784, 882 and 189. Please address this difference and the choice of CHF1 over CHF33 given this phenotypic data.
  • Figure 2A: The plaques in the figure show varying sizes, is this representative of the purified phage activity? The authors mention this is representative of the plaque morphology of all phages – did all contain a combination of large and small resulting plaques?
  • Line 250-252: When describing the sequencing data, it would be beneficial to provide context of the overall findings including genome sizes and sequence coverage of the assemblies. I note additional information is available in Supplementary Table 2, but this is worth noting in text also.
  • Figure 3: I suggest this figure be made using Easyfig (https://mjsull.github.io/Easyfig/), a free software that enables visual representations of alignments. This would enhance the reader’s comprehension of the identity across all phages.
  • Line 270-306: This section of results needs to be thoroughly revised as it is difficult to understand the results that are detailed. Firstly, the use of multiple phages is referred to as a cocktail rather than a “mix” and use of this term would be easier to follow in the text and figures. Similarly, in places where different numbers of phages have been combined (e.g. Fig 1B = x4 and Fig 5A = x2), detail of specific phages tested is required. The data is difficult to interpret in its current labelled form and needs to be reassessed:
    • Psa+phages: Which phages? Individual phages averaged or all phages together?
    • Control: Is this without phage and without Psa? A negative control? If so, why not label “Psa” as a positive control?
  • Spelling and grammatical:
    • Line 46-47: Replace “…there are six biovars of Psa, being the biovar 3 responsible…” with “…there are six biovars of Psa, of which biovar 3 is responsible…”.
    • Line 48: Replace “exports in near 55%...” with “exports by approximately 55%...”.
    • Line 65: Replace “In vitro” with “in vitro” (italicised).
    • Line 75-78: Please clarify the purpose of the strains listed as you have done for the phage isolation strains. For example “Specificity was tested against commensal kiwifruit bacterial isolates including Pseudomonas putida…”.
    • Line 78, 81, 134, 278, 279, Supp. Table 2, etc.: Several places where comma is used rather than period or vice versa. Please check this throughout the manuscript.
    • Line 102: Replace “To phage…” with “The phage…”.
    • Line 162: Remove “than were”.
    • Line 184: Replace “…plants no infected with Psa” with “…plants not infected with Psa”.
    • Line 191-192: Rearrange the sentence to read along the lines of “The phages were isolated using different strains of Psa biovar 3 obtained from Chilean kiwifruit orchards as the hosts”.
    • Line 246: Replace “D.O” with “OD”.
    • Line 199: Rearrange the sentence to read along the lines of “…included in this study, with phages CHF21, CHF30 and CHF33, observed with the greatest lytic activity against all isolates…”.
    • Line 231: Replace “loans” with “lawns”.
    • Line 235: Replace “pb” with “bp”, this typographical error appears in several places in the manuscript, please correct throughout.
    • Line 303: Replace “…discs no infected with Psa” with “…discs not infected with Psa”.
    • Line 321: Replace “…leaves in more than 75%...” with “…leaves by more than 75%...”.
    • Line 325: Replace “produce” with “produced”.

Author Response

We thank the constructive and thoughtful comments of reviewer 1

Specific comments:

-Line 116: Please name Restriction Fragment Length Polymorphism (RFLP) in full before referring to the acronym.

The name Restriction Fragment Length Polymorphism was included in Line 120

-Figure 1: There are a few corrections that would improve the readability of this figure. Firstly, the in-figure legend should read in the same order as the columns are presented as in panel A. Panel B and C are written in the reverse order with control last. In addition, it would be beneficial to use a more discerning colour scheme to easily visualise the different conditions.

The figure 1 was modified following the suggestions of the reviewer.

-Line 219, Figure 1 description: Reword this for clarity to something like “…phages were exposed to pH 4 – 6 for 1 hour (A), temperature of 4°, 18° and 37°C for 1 hour and solar…”.

The description of figure 1 was modified following the suggestions of the reviewer.

Line 232 “The fourteen phages were exposed to pH 4–6 for 1 hour (A), temperatures of 4, 18, and 37 °C for 1 hour (B) and solar radiation (UV level 8) for 30 or 60 min (C).”

-Line 238-240: It is interesting that of the near identical phages CHF1 and CHF33 that the latter (CHF33) was excluded from further experiments given that the host range data suggests greater lytic activity compared to CHF1 against Psa hosts 598, 784, 882 and 189. Please address this difference and the choice of CHF1 over CHF33 given this phenotypic data.

We agree with the reviewer that phage CHF33 shows a broader host range than the phage CHF1; however, this was not considered because the combination of phages F1, F7, F19 and F21 allows an efficient infection in all the Psa isolates tested. We chose CHF1 over CHF33 because we were regularly obtaining better titters for this phage. For better clarification, we have included the following sentence in the manuscript.

Line 256 “The phage CHF33 showed stronger lytic activity over different Psa isolates in comparison to CHF1 (18 vs. 14 isolates); however, the titers obtained during phage production were consistently higher in CHF1 and CHF19 than CHF33 and CHF17. Therefore, the latter phages were excluded from further experiments. The combination of the selected phages CHF1, CHF7, CHF19, and CHF21 demonstrated efficient infections in all the Psa isolates tested in the study (Supplementary Table 1).”

-Figure 2A: The plaques in the figure show varying sizes, is this representative of the purified phage activity? The authors mention this is representative of the plaque morphology of all phages – did all contain a combination of large and small resulting plaques?

We thank the question of the reviewer. Indeed the phages showed mostly large plaques, but occasionally, some small plaques were observed. We purified small and large plaques to determine if there were contamination, but we observed the same phenomena; therefore, we assume the small plaques correspond to phage variants that emerge spontaneously. For major clarification, we included the following sentence in the text.

Line 245 “All the selected phages produced large clear lytic plaques in Psa lawns, but occasionally small size plaques were also observed (Figure 2A)”

-Line 250-252: When describing the sequencing data, it would be beneficial to provide context of the overall findings including genome sizes and sequence coverage of the assemblies. I note additional information is available in Supplementary Table 2, but this is worth noting in text also.

Most of the information included in Supplementary Table 2 (Genome size, %GC, number of ORFs and %identity with phage phiPsa2) is already included in the text. In order to provide additional information, the following sentence was added to the text:

Line 249 “The genome of the selected phages was sequenced using NGS with a sequence coverage above 1000 for each phage. The phages have a genome size ranging between 40,557 and 40,999 bp”

-Figure 3: I suggest this figure be made using Easyfig (https://mjsull.github.io/Easyfig/), a free software that enables visual representations of alignments. This would enhance the reader’s comprehension of the identity across all phages.

Following the recommendation of reviewer 1 we have included a visual representations of genome phage alignments elaborated by Easyfig software.

-Line 270-306: This section of results needs to be thoroughly revised as it is difficult to understand the results that are detailed. Firstly, the use of multiple phages is referred to as a cocktail rather than a “mix” and use of this term would be easier to follow in the text and figures. Similarly, in places where different numbers of phages have been combined (e.g. Fig 1B = x4 and Fig 5A = x2), detail of specific phages tested is required. The data is difficult to interpret in its current labelled form and needs to be reassessed:

-Psa+phages: Which phages? Individual phages averaged or all phages together?

-Control: Is this without phage and without Psa? A negative control? If so, why not label “Psa” as a positive control?

According to the suggestions of the reviewer the full section was revised and edited. The figures and associated text were also edited according to the suggestions. All the changes were made with Track changes and can be found in the new version of the manuscript. Besides the entire text was subjected to English edition by an English edition services provided by MDPI.

The final version of the section commented by the reviewer reads as follows:

Line 293 “The lytic activity of the phages CHF1, CHF7, CHF19, and CHF21 was evaluated through infection curves using the respective Psa host for each phage. The results show that all phages were able to clear the bacterial culture, even when a MOI of 0.1 was used (Figure 2C). The phage CHF1 was the most effective at lysing the bacterial culture within 2 hours, while the phage CHF19 needed up to 6 hours to clear the Psa culture. Considering the potential use of these phages as biocontrol agents against Psa, the frequency of bacteriophage resistant mutants was evaluated. The phages CHF1, CHF7, CHF19, and CHF21 selected for bacteriophage resistant mutants with a frequency of 4.79 x 10-6, 9.05 x 10-6, 9.3 x 10-6, and 3.28 x 10-6, respectively. In all cases, three different colony morphologies were observed (normal Psa-like, small, and smooth borders (data not shown)).

The potential of these phages as biocontrol agents for Psa was first evaluated under laboratory conditions with kiwifruit leaf samples. Our results show that a cocktail of phages CHF1, CHF7, CHF19, and CHF21 in equal proportion was able to reduce the bacterial load of Psa over leaves below our detection limit (20 UFC/mL) within 3 hours post-infection (hpi), and remained undetected up to 24 hpi (Figure 4A). During the same period, the Psa load in the untreated leaves ranged between 105–107 UFC/mL. Bacteriophages were detectable throughout the entire experiment, showing an increase in their titer at 3 hours, probably due to their replication in Psa (Figure 4A).

To evaluate if the cocktail of phages was able to protect kiwifruit leaves against the damage produced by Psa, an in vitro methodology proposed by Prencipe et al., (2018) [41] was implemented. The results show that these phages were also able to protect kiwifruit leaf discs from the damage produced by Psa. After three days, leaf discs that received the phage cocktail treatment presented an average damage index of 1.5, while the untreated discs presented a damage index of 2.9. Multiple doses of phage cocktail resulted in a damage index of 0.8, showing significant differences with the untreated discs (Figure 4B). No damage was observed in the control leaf disc groups that were not infected with Psa or treated with phages. These results suggest that this cocktail of bacteriophages had the potential to control the infection produced by Psa in kiwifruit tissue, reducing the bacterial load and symptomatology in leaves.

Spelling and grammatical:

  • Line 46-47: Replace “…there are six biovars of Psa, being the biovar 3 responsible…” with “…there are six biovars of Psa, of which biovar 3 is responsible…”.
  • Line 48: Replace “exports in near 55%...” with “exports by approximately 55%...”.
  • Line 65: Replace “In vitro” with “in vitro” (italicised).
  • Line 75-78: Please clarify the purpose of the strains listed as you have done for the phage isolation strains. For example “Specificity was tested against commensal kiwifruit bacterial isolates including Pseudomonas putida…”.
  • Line 78, 81, 134, 278, 279, Supp. Table 2, etc.: Several places where comma is used rather than period or vice versa. Please check this throughout the manuscript.
  • Line 102: Replace “To phage…” with “The phage…”.
  • Line 162: Remove “than were”.
  • Line 184: Replace “…plants no infected with Psa” with “…plants not infected with Psa”.
  • Line 191-192: Rearrange the sentence to read along the lines of “The phages were isolated using different strains of Psa biovar 3 obtained from Chilean kiwifruit orchards as the hosts”.
  • Line 246: Replace “D.O” with “OD”.
  • Line 199: Rearrange the sentence to read along the lines of “…included in this study, with phages CHF21, CHF30 and CHF33, observed with the greatest lytic activity against all isolates…”.
  • Line 231: Replace “loans” with “lawns”.
  • Line 235: Replace “pb” with “bp”, this typographical error appears in several places in the manuscript, please correct throughout.
  • Line 303: Replace “…discs no infected with Psa” with “…discs not infected with Psa”.
  • Line 321: Replace “…leaves in more than 75%...” with “…leaves by more than 75%...”.
  • Line 325: Replace “produce” with “produced”.

All the suggestions made by the reviewer were included in the manuscript. Besides the complete text was revised and edited by an English editing service provided by MDPI.

Reviewer 2 Report

Flores et al., characterize 14 phages and 4 of them resulted able to reduce the Psa bacterial load. The manuscript is well written, the material and methods is adequate and the discussion is appropriate. This work is worthy of publication; however, two queries need some clarification:

  • In Figure 1A, the Ph seem to compromise the phages and the CHF15 has showed an increment higher than the titre of the Control. How this could be explained?

  • Only the phage CHF1 contains an extra “0”. Authors should have re-amplified this ORF and re-sequenced it to ascertain its real existence and that it is not an artifact of the Illumina contigs assembling.

  • There was any relationship between the RFLP, and sequence variations found, for all identified bacteriophages? No information about that in the text.

Author Response

We thank the constructive and thoughtful comments of reviewer 2

-In Figure 1A, the Ph seem to compromise the phages and the CHF15 has showed an increment higher than the titre of the Control. How this could be explained?

Our results suggest that these phages are differentially affected by pH. Some of them, such as CHF15 or CHF30 seem to be unaffected at pH 6. According to our results, the differences observed for phage CHF15 between the control condition and pH6 are not significant (p<0.001), therefore to our understanding, these differences correspond to the variability of the double-layer agar assay.

-Only the phage CHF1 contains an extra “0”. Authors should have re-amplified this ORF and re-sequenced it to ascertain its real existence and that it is not an artifact of the Illumina contigs assembling.

We thank Reviewer 2 for the comments and the suggestions. However, at this moment, it would be extremely difficult to perform experimental procedures at the laboratory. Our University is closed to in situ activities due to the global pandemic. Moreover, our group is currently collaborating with the COVID-19 sample analysis; therefore, we’re not in the condition to confirm the presence of ORF 0 in phage CHF1 experimentally. However, we’re very confident that this ORF is not an artifact of the Illumina contigs assembling due to the following reasons.

After sequencing, we obtained several millions of paired-end reads (100 bp) for each phage, which assure a high level of coverage for genomes of 40 kb. To reduce errors in the assembling, reads were processed for removal of adapters, duplicates and quality trimming (Q >30). All genomes were assembled using a de novo strategy using the Geneious R11 software, obtaining unique contings for each phage, with a coverage of 17,489x for phage CHF1. The same result was obtained using the CLC genomics software. The genomes were also assembled to the reference using the genome of the phage phiPsa2, which share high identity with the phages sequenced in this study (90%<). In all cases, the phage CHF1 presents the extra ORF 0. Finally, this same ORF 0 was present in phage CHF33, which shares a 99.96% identity with CHF1, and was isolated, sequenced, and assembled separately. We believe that this information strongly suggests that the ORF0 is not an artifact of the assembly.

For better clarification, the following sentences were included in the text.

Line 128 “The phages were assembled following a de novo strategy and the results were then confirmed as assembled to the reference using the genome of the phage phiPSA2.”

Line 277 “Only the phages CHF1 and CHF33 contained an extra ORF (ORF 0), which was annotated as a hypothetical protein”

-There was any relationship between the RFLP, and sequence variations found, for all identified bacteriophages? No information about that in the text.

According to our analysis, there is a correlation between RFLP and the genomic analysis. The restriction patterns are more similar between phages CHF1 with CHF33, and CHF17 with CHF19 than with other phages. However, it is important to note that the differences observed by RFLP analysis are minimal (only a few bands). Besides, RFLP analysis is much less resolutive than genomic analysis. For better clarification, we have included the PyElph analysis in the Supplementary figure 1, and the following sentence in the text.

Line 395 “The sequenced genomes revealed that these phages shared nucleotide identities over 96% among them, with two couples (CHF1 and CHF33; and CHF17 and CHF19) that were essentially identical (>99.9% identity). This information correlates with the observations in the RFLP analysis (Supplementary Figure 1), where CHF1 share more similitude with CHF33 and CHF17 with CHF19”.

All the changes in the manuscript are highlighted with Track changes

Round 2

Reviewer 1 Report

The authors have responded to the review comments and adjusted the manuscript accordingly.

Reviewer 2 Report

No comments